# SARS-CoV-2 Omicron: Viral Evolution, Immune Evasion, and Alternative Durable Therapeutic Strategies

**DOI:** 10.3390/v16050697

**Published:** 2024-04-28

**Authors:** Hailong Guo, Sha Ha, Jason W. Botten, Kai Xu, Ningyan Zhang, Zhiqiang An, William R. Strohl, John W. Shiver, Tong-Ming Fu

**Affiliations:** 1IGM Biosciences, Mountain View, CA 94043, USA; 2Department of Medicine, Division of Pulmonary Disease and Critical Care Medicine, Robert Larner, M.D. College of Medicine, University of Vermont, Burlington, VT 05405, USA; 3Department of Microbiology and Molecular Genetics, Robert Larner, M.D. College of Medicine, University of Vermont, Burlington, VT 05405, USA; 4Texas Therapeutics Institute, Brown Foundation Institute of Molecular Medicine, The University of Texas Health Science Center at Houston, Houston, TX 77030, USA

**Keywords:** SARS-CoV-2, evolution, ACE2, therapy

## Abstract

Since the SARS-CoV-2 Omicron virus has gained dominance worldwide, its continual evolution with unpredictable mutations and patterns has revoked all authorized immunotherapeutics. Rapid viral evolution has also necessitated several rounds of vaccine updates in order to provide adequate immune protection. It remains imperative to understand how Omicron evolves into different subvariants and causes immune escape as this could help reevaluate the current intervention strategies mostly implemented in the clinics as emergency measures to counter the pandemic and, importantly, develop new solutions. Here, we provide a review focusing on the major events of Omicron viral evolution, including the features of spike mutation that lead to immune evasion against monoclonal antibody (mAb) therapy and vaccination, and suggest alternative durable options such as the ACE2-based experimental therapies superior to mAbs to address this unprecedented evolution of Omicron virus. In addition, this type of unique ACE2-based virus-trapping molecules can counter all zoonotic SARS coronaviruses, either from unknown animal hosts or from established wild-life reservoirs of SARS-CoV-2, and even seasonal alpha coronavirus NL63 that depends on human ACE2 for infection.

## 1. Introduction

SARS-CoV-2 virus has caused about 772 million confirmed infections and over 6.9 million death toll globally reported to WHO as of 8 December 2023, and it has been a continual public health threat during and post-pandemic phases due to high viral evolution. Initial variants such as alpha, beta, gamma, and delta that appeared in 2020 had dominated for most of 2021 globally, concurred with increased transmission and infectivity, mostly due to infection in relatively naïve populations. The monoclonal antibodies (mAbs) approved under emergency use authorization (EUA) and vaccines developed using the original Wuhan strain retained reasonable activities against these variants [1,2,3]. In contrast, the constellation of Omicron variants, initially reported in the fall of 2021, has shown an evolution pattern with not only superior transmission and infection but also evasion of EUA-approved mAbs and immunity induced by vaccination or natural infection.

## 2. Emergence of Omicron

In late 2021, the Omicron variant rose and replaced the Delta virus due to its much higher infectivity and transmission efficiency than all previous variants [4,5]. Although disease severity caused by the Omicron variants is generally lower compared to the Delta virus [6], the resulting outbreak still caused about 60,000–100,000 weekly mortalities worldwide between December 2021 and March 2022, according to WHO’s COVID19 Dashboard (accessed on 10 December 2023), and the highest hospital admissions during entire pandemic phases. It should be noted that, during the Omicron outbreak, certain populations, such as cancer patients and the elderly, regardless of vaccination status, continually experienced a significantly higher fatality rate [7,8,9], and school-aged children and infants were seen with increased hospital admissions and fatalities [8,10,11]. The most striking feature of the Omicron virus is its capability of diverse within-lineage evolution and divergent cross-lineage recombination [12,13] that are distinct from the relatively slow and adaptive substitutions of early variants, and its striking manifestation of continual immune evasions due to increased mutations on spike proteins [14,15]. The unusual long branch of the Omicron variant in the constructed evolutionary tree (Figure 1) has suggested a different and perhaps mysterious evolutionary origin and trajectory as indicated by several proposed theories, such as human and mouse interspecies evolution [16,17].

## 3. Omicron Sublineages and Immune Evasion

### 3.1. BA.1, BA.2 and BA.3

Soon after the initial debut in South Africa, the Omicron virus had rapidly evolved within a few months into three lineages, including BA.1, BA.2, and BA.3, dominated briefly by BA.1 and then quickly replaced by BA.2 while BA.3 had caused a limited number of cases [18] (Figure 1). All these Omicron variants harbor about 30 mutated amino acids in the spike protein, triple that seen in alpha, beta, and gamma variants (Figure 2). BA.1, BA.2, and BA.3 share a total of 21 spike mutations with unique ones identified on BA.1 and BA.2 but not BA.3 [19]. It was found that not all mutations on the spike proteins of BA.1, BA.2, and BA.3 viruses are beneficial for viral replication and escape. However, the compensated effects of these heavily mutated residues on these Omicron variants enhance ACE2 binding, viral fitness, transmission, and alter antigenicity [20,21]. These extensive mutations are located on the viral spike surface area, which is critical for the interaction of the spike receptor binding domain (RBD) with the angiotensin-converting enzyme 2 (ACE2) receptor, and on several well-characterized antigenic sites [20,22]. Therefore, in contrast to the early variants, these Omicron subvariants demonstrated their abilities to escape a large portion of mAbs in preclinical evaluation and clinical development [23,24,25]. In addition, sera from individuals who tested positive for prior infections with early variants or underwent single or full-dose vaccinations with mRNA or protein vaccine based on WT sequence dramatically lost neutralization efficacies against these new variants [24,25,26]. Due to the overall shared sequences and structures, it is not surprising to see that the sera from BA.1 infection can cross-neutralize against BA.2 and BA.3, albeit with a few-fold lower potency than BA.1 [27]. However, BA.2 has several unique mutations that could contribute to antigenic difference, enhanced resistance to mAb and sera neutralization, and reinfection with BA.2 after BA.1 [28,29,30].

### 3.2. BA.2.12.1, BA.4 and BA.5

The BA.2 virus itself has subsequently evolved into other different sublineages such as BA.2.12.1 (Figure 2), which was first detected on February 2022 in Australia, Luxembourg, and Canada based on earliest available sequences deposited in GISAID (EPI_ISL_9767878, EPI_ISL_9800916, EPI_ISL_9851031, Accessed on October 3, 2023). Almost at the same time, BA.4 and BA.5 viruses were identified in South Africa, ousting the dominant BA.2 in April, spreading and causing an additional wave of worldwide outbreaks [33,34]. These BA.2 sublineages have a better effective reproduction number, Rt, and infect more efficiently than the original BA.2 [35], which contributed to the global replacement of BA.2 circulation. BA.4 and BA.5 have differences in the C terminal region of the genome but share the same spike mutations that differ from BA.2 by a deletion of 69–70, two mutations of L452R and F486V, and a reversion of R493Q [33]. The L452 mutation (L452Q) is also displayed in BA.2.12.1 in addition to S704L. The individual mutation of L452 and F486 or the combination of both have contributed to the escape of these new Omicron sublineages (BA.2.12.1, BA.4, and BA.5) from all classes of antibodies directed against the spike RBD except the clinically approved drug: Bebtelovimab [36], (Figure 2). Further, these new mutations are attributed to the capabilities of BA.2.12.1, BA.4, and BA.5 to transmit and evade host immunity to the point that even sera from BA.1/BA.2 infection or vaccination with the then updated bivalent vaccine containing original and BA.1 spike mRNA were no longer effective [27,35,37,38,39], and breakthrough infections were routinely reported [29,40]. Although some studies showed that breakthrough infections with early variants might help improve immunity [41,42], when double or booster-vaccinated patients encountered BA.5 reinfection, their immune responses were neither enhanced [43] nor protective against infection with other variants [44]. Further, re-infection with BA.5 could reduce the neutralization efficacy against new variants [45,46], likely reflecting the original antigenic sin responses as recently demonstrated for SARS-CoV-2 and previously for influenza viruses [47,48,49], or the poor immunogenicity of BA.5 during reinfection as observed for other Omicron viruses [50].

### 3.3. BA.2.75

Just as the BA.4/BA.5 wave started to peak, a new sublineage of BA.2, named BA.2.75, was first sequenced in India in early June of 2022 and then detected in most other countries worldwide, with cases peaking from October to November [51,52] (Figure 1). Compared to BA.2 and BA.4/BA.5, this sublineage acquired several new mutations, including G446N and N460K [53], which contribute to its enhanced immune evasion [54,55]. Further, sera from BA.5 breakthrough infections showed weaker neutralization against BA.2.75 than BA.5 [54] and BA.5 bivalent vaccination was not able to elicit high neutralization response to BA.2.75 [56]. Although the reversed R493Q mutation could potentially reduce BA.2.75 neutralization resistance, all the mAbs except Bebtelovimab and a combination of Tixagevimab and Cilgavimab lost efficacy [54] (Figure 2). However, with further evolution and spike mutations, such as R346T and F486S identified in its progeny, BA.2.75.2, the neutralization potency of Tixagevimab and Cilgavimab was also diminished [57,58].

### 3.4. BQ.1 and XBB

Although it was predicted with super transmission capability, BA.2.75 had only spread transiently in most countries and failed to completely outcompete other sublineages according to GISAID’s hCoV-19 Variants Dashboard (Reviewed on 6 October 2023) as new Omicron lineages were bursting out during its peak phase of detection. One of the noticed variants was BQ.1, which was detected from samples collected in Nigeria and the US in July 2022, followed by its progeny BQ.1.1 (GISAID, Figure 2). BQ.1 was characterized as a derivative of BA.5 (Figure 1) with two additional spike mutations (K444T and N460K) for BQ.1 and three (K444T, N460K, and R346T) for BQ.1.1 [51]. Then in a matter of another month, XBB and XBB.1 were identified as recombinants of two BA.2 sublineages, BA.2.10.1 and BA.2.75, first found in India and then globally [59]. The XBB sublineage has 14 spike mutations (9 in RBD) in addition to those found in BA.2, whereas XBB.1 has one additional G252V mutation [59]. These newly rising sublineages showed similar fitness advantages and were cocirculating with shared spike mutations that were convergently acquired during Omicron evolution [60,61]. Compared to BA.5, BQ.1.1 evades breakthrough BA.2 and BA.5 infection sera more efficiently, and sera from BA.5 bivalent vaccination only showed low-level neutralization activity toward BQ.1.1 and XBB.1 viruses [56,60], indicating enhanced immune evasion of these cocirculating sublineages.

BQ.1, BQ.1.1, XBB, and XBB.1 were considered the most resistant subvariants against known mAbs, as even Bebtelovimab, the broadest authorized anti-COVID antibody, was no longer effective against these variants [51,62,63,64]. In addition, Evusheld, the combination of Tixagevimab and Cilgavimab, also failed to neutralize BQ.1 and XBB-derived viruses [62,63,65] (Figure 2). Such alarming evolution and immune evasion have also been demonstrated for other Omicron sublineages, such as BA.4.6.3 and CH.1.1, but these had spread less widely than BQ.1 and XBB viruses [63]. Mechanically, the super potent immune resistance of these viruses is conferred via the convergently mutated resides in the spike protein that can disrupt hydrogen binding or form steric clashes with the mAbs [51,63]. More importantly, repeated vaccine boosters and breakthrough Omicron infections could lead to narrowed neutralization epitope repertoires and increase the proportion of non-neutralization fractions targeting WT RBD epitopes [63]. This original antigenic sin or immune imprinting might further promote Omicron virus evolution and escape [63,66,67].

### 3.5. XBB.1 Derivatives

XBB.1 virus quickly underwent additional mutations, drifting into different sublineages such as XBB.1.5 and XBB.1.16 (Figure 2) that have cocirculated since late 2022 with other subvariants, showing enhanced evasion of antibody neutralization but not disease severity [68,69,70]. While XBB.1 sublineages were spreading globally, a new sublineage EG.5 descended from XBB.1.9.2 was detected in February 2023 with additional spike mutations such as F456L, one of the key FLip mutations (L455F and F456L) that were predicted via deep mutational scans [71,72]. EG.5 was classified as a new variant of interest on 9 August 2023, by WHO (Weekly epidemiological update on COVID-19 -10 August 2023) and has been increasingly detected in the US and some other countries along with its derivatives, although some scientists had initially predicted its low potential of widespread [73]. As expected, due to the FLip mutations located on the spike RBD, EG.5 and some other latest XBB.1 variants, including XBB.1.16 and XBB.2.3 (Figure 2), were substantially more resistant to neutralization by serum collected from individuals who received the BA.5 bivalent booster or experienced breakthrough infections with BQ or early XBB sublineage [74,75,76].

### 3.6. BA.2.86 and JN.1

BA.2.86 is the latest sublineage, which emerged from Omicron BA.2 that was initially detected in Europe, Israel, and the US [77]. BA.2.86 has been considered to be the most astonishing sublineage since the start of Omicron [77,78], as it carries more than 30 spike protein mutations compared with its parental BA.2 [79,80] (Figure 2). And soon after the discovery, it was designated as variant under monitoring. The preliminary assessment of Rt suggested this virus could have better fitness than currently circulating XBB variants, including EG.5.1 [80]. All previous EAU mAbs did not show robust antiviral inhibition against BA.2.86 [79,80], nor the sera obtained from individuals administered with multiple doses of monovalent or BA.1 and BA.5 bivalent mRNA vaccines [80]. Further, BA.2.86 appears slightly more or at least similarly resistant to serum neutralization when compared to XBB.1.5 [79,81], and neutralization of XBB breakthrough infection sera against this new virus is significantly lower than that against EG.5.1 [80]. However, this subvariant was not able to compete with other circulating strains to become dominant until the appearance of an additional spike mutation, L455S, forming a new derivative, named as JN.1 [82]. Currently, JN.1 has become prevalent in Europe [83] and the US according to CDC’s variant tracking data.

### 3.7. Summary of Omicron Variants

The astonishing evolutions of Omicron viruses have pushed the withdrawal of all EUA mAbs from clinical usages and the updating of the COVID-19 vaccine to include the coverage of new sublineages several times already. It is possible that new mAbs could still be developed with specificity and sufficient potency against newly emerged Omicron viruses. However, based on what we have experienced and learned, these biologics are unlikely to sustain their efficacies against future SARS-CoV-2 variants and sublineages. Alternative durable therapeutic approaches are much needed to overcome the challenge of rapid Omicron evolution, possible retransmission of CoV-2 virus from established animal reservoirs where it has undergone its own evolution [84], and new spillover of other alpha and beta CoVs from bats and/or pangolins.

## 4. ACE2 Based Therapeutic Approaches

ACE2 was originally discovered as a homolog of angiotensin-converting enzyme (ACE), but it counteracts ACE-mediated detrimental pathway by degrading angiotensin II to negatively regulate the renin-angiotensin system (RAS) for maintaining blood pressure homeostasis and protecting major organs from injuries [85,86]. Soon after the outbreak of SARS-CoV (now known as SARS-CoV-1) in 2003, ACE2 was identified as its functional receptor [87], which is recognized by viral spike RBD [88]. Due to the high sequence homology of the SARS-CoV-2 spike with that of SARS-CoV-1, ACE2 was proposed as a receptor for SARS-CoV-2 and soon thereafter confirmed experimentally [89,90] with detailed structural characterizations [91,92].

Although SARS-CoV-2 viruses have undergone extensive evolution, causing several waves of outbreaks, the variants and sublineages that emerged still use ACE2 as a functional receptor for infection, and blockage with soluble ACE2 can effectively inhibit their infections better than WT viruses [93,94]. The similar or enhanced bindings and interactions with ACE2, fusions, and syncytia formations by RBD of all major variants and Omicron sublineages [35,39,95,96,97,98,99,100] have also supported the fundamental concept that efficient viral entry and productive infection are required for successful transmission and spreading of these variants. So far, there is no indication that SARS-CoV-2 replication deviates away from the dependence of ACE2. Further, ACE2 is flexible enough to tolerate the diverse evolution of different CoV genera, including alpha and beta, which have distinctive receptor motifs on their RBDs with limited homology used for interaction and binding with ACE2, as shown in the structural footprint analysis (Figure 3). The human alpha CoV, NL63, was likely derived from the recombination of early versions of NL63-like and 229E-like viruses circulating in the bats a century ago [101], suggesting the ancestral nature of alphavirus RBD binding with ACE2. We performed structural modeling and analysis using SWISS-MODEL and PyMOL, and identified a potential ACE2 binding motif similar to human NL63 for a recent 229E-related bat virus isolate (5425) (NCBI GenBank: MN611517.1). We also have data showing that the unmodified RBD of 5425 virus could directly interact with human ACE2 with an Octet binding IC_50_ of <1 pM similar to the NL63 RBD [102], indicating that alpha CoV RBD footprint on ACE2 (Figure 3) is well conserved and still functionally maintained in wild animals today. It is likely that RBD binding of beta CoVs with ACE2 is also evolutionally stable for at least some animal isolates that have a high potential of infecting humans [103,104,105]. Therefore, theoretically, targeting ACE2 for developing therapeutic biologics could provide an evolutionarily durable solution for addressing the ever-increasing SARS-CoV-2 mutations and future pandemic threats. Currently, several strategies have already been evaluated in preclinical and clinical studies. Below, we discuss three major advances, including ACE2 peptide mimic, anti-ACE2 mAb, and ACE2 trap.

### 4.1. ACE2 Peptide Mimic

The full-length extracellular domain of ACE2 contains about 740aa. However, the main spike RBD interaction sites are located on its small N-terminal alpha helices that are uncoupled from the catalytic function [88,106]. Therefore, it was proposed that small molecules blocking the RBD-ACE2 interaction could be developed as anti-SARS-CoV drugs [107]. Several groups have tried to design small peptides or ACE2 mimics based on the structure of the RBD-interacting alpha helix. However, the derived peptides usually show low RBD binding and poor viral inhibition for SARS-CoV viruses. For example, the ACE2 peptides aa22-44 and aa22-57 only exhibited modest antiviral activity with IC_50_ over 5–50µM for SARS-CoV-1 [108], and the best peptide designed by RC Larue et al. showed only about 0.5 mM IC_50_ for binding SARS-CoV-2 and 2.0 mM IC_50_ for inhibiting SARS-CoV-2 infection [109]. Similarly, the best ACE2 peptide candidate designed by P Adhikary et al. could inhibit SARS-CoV-2 infection but with over 1.0 mM IC_50_ in a neutralization assay [110]. New efforts were attempted, including de novo design with or without relying on ACE2-RBD structural interactions. Using this approach, L Cao et al. obtained a few candidates after screening two sets of peptide libraries with enhanced RBD binding affinity up to 0.1 nM and viral inhibition IC_50_ of 24 pM [111]. One of the modified lead candidates, LCB1v1.3, was shown to provide protection in vivo against the challenge of both WT and alpha variant SARS-CoV-2 in K18-hACE2 mice [112]. Further study, however, showed additional mutations on emerged variants, such as Beta and Gamma, would significantly impact its potency [113], raising concern about the susceptibility of this type of artificially designed candidate to SARS-CoV-2 evolution.

### 4.2. Anti-ACE2 mAb

Because RBD-ACE2 interaction is required for SARS-CoV-2 entrance and infection, blocking this interaction should, in theory, prevent infection. Spike-targeting mAbs, especially those directed against RBD, have been proven to be able to potently interfere with ACE2 interaction and neutralize virus. However, as discussed, these mAbs could not resist viral evolution, preventing them from continual clinical usage. Alternatively, several investigators have started to develop mAbs against ACE2 instead of RBD. For example, M Hoffmann et al. showed that polyclonal anti-ACE2 could inhibit both SARS-CoV-1 and SARS-CoV-2 pseudoviruses in vitro with an estimated IC_50_ around 2 µg/mL [114]. Y Du et al. generated a humanized anti-ACE2 mAb, h11B11, that could block RBD binding to ACE2 [115]. In vitro, neutralization showed an IC50 of about 1 µg/mL for the WT SARS-CoV-2 virus, which was similar in potency to that of the anti-ACE2 polyclonal antibody described by M Hoffmann et al. [114]. With single RBD mutations such as V367F and N354K, this mAb lost several folds of anti-viral potency. In vivo, treatment with a high dose (25 mg/kg) after the WT virus challenge showed only minor inhibition of viral replication [115]. Therefore, both in vitro and in vivo data indicated that this anti-ACE2 mAb is not potent for viral neutralization and clearance. Similarly, Y Chen et al. developed a mouse hybridoma-derived anti-ACE2 mAb, 3E8, which showed modest binding affinity to ACE2, in vitro viral neutralization, and in vivo reduction of viral replication with prophylactic treatment [116]. AE Chaouat et al. also generated nine anti-ACE2 mAbs through mouse immunization and screening. However, only one (hACE2.16) could block RBD-ACE2 interaction without affecting ACE2 enzymatic activity [117]. Further, in vitro, virus neutralization required a high IC_50_ ranging from 2 to 8 µg/mL for WT, Alpha, Beta, Gamma, and Delta variants, although some improvement was observed for Omicron BA.1 (0.96 µg/mL) and BA.2 (0.26 µg/mL). No in vivo assessment was provided [117]. Most recently, F Zhang et al. generated a panel of anti-ACE2 mAbs with improved neutralization potencies of less than 10 ng/mL against WT, early variants, and Omicron BA.1 [118]. Prophylactic treatment of hACE2 knock-in mice with 12.5 mg/kg of these anti-ACE2 mAbs could inhibit replication of WT SARS-CoV-2 in the lung. However, the authors did not provide data on clinical and pathological outcomes and assessment of recent Omicron sublineages. Altogether, although these studies have indicated the possibility of developing an anti-ACE2 mAb for treating COVID-19 infections, broad coverage against diversified Omicron evolutions has not been demonstrated, and further safety evaluation will be needed for such a molecule that directly targets ACE2 as physiologically it is a master regulator for blood pressure homeostasis and tissue protection [85,86,119,120].

### 4.3. ACE2 Trap

The use of viral receptors for viral trapping has been applied to several viruses, including HIV-1, hepatitis A virus (HAV), and rhinovirus [121,122,123]. This strategy has also been explored for SARS-CoV-2. One early method is using an engineered soluble form of ACE2 (sACE2) with (AA18-740) or without (AA18-615) the neck domain (Table 1). These sACE2 traps can inhibit SARS-CoV-2 attachment and entry into host cells and exhibit antiviral activities against multiple variants. Generally, however, the potency of sACE2 is low [124,125,126,127,128,129]. One of the most developed sACE2 molecules, APN01, is a soluble monomeric protein containing the ACE2 neck domain (AA18-740). It was shown to have the ability to block infection in human tissue organoids and neutralize the WT virus and the early variants, including Alpha, Beta, and Gamma [127,130]. However, when given intravenously (IV) to severe COVID-19 patients, it did not significantly prevent death, reduce days on ventilator, or shorten the time for hospital discharge (NCT04335136), possibly due to low antiviral activity and fast clearance of this soluble recombinant candidate in patients. It has also been evaluated as an inhalation formulation and found to be safe in toxicology studies in dogs [131], with a Phase 1 clinical study (NCT05065645) completed with results not posted yet.

Two major methods have been attempted to increase the potency of the ACE2 trap. One is called affinity maturation, which relies on mutating specific residues in ACE2 N-terminal alpha helices or surrounding regions to promote RBD binding and antiviral potency (Table 1). For example, when a native form of sACE2 was engineered with three mutations, T27Y, L79T, and N330Y, its neutralization was improved about 10-fold with an IC_50_ around 60 ng/mL for WT virus [124]. To enhance potency and serum half-life, sACE2 has been frequently fused with IgG Fc, resulting in bivalent ACE2 that can be administered less frequently than sACE2 [124,125,126,129,132,134,137,140,141]. It appears, however, that these bivalent ACE2s in the native form are still low in neutralization potency, as shown by the IC_50_ for WT virus that ranges from 400–24,800 ng/mL (Table 1). For further improvement, affinity maturation was also applied to some of these molecules. One example is 3N39v2 [137], an ACE2 without the neck domain that contains affinity mutations of A25V, K31N, E35K, and L79F that lead to about 300-fold improvement in neutralization IC_50_, reaching 82 ng/mL for WT virus (Table 1). Further optimization and inclusion of the ACE2 neck domain in a follow-up study led to candidate 3N39v4 [138]. This lead did not appear to result in much improvement for neutralization IC_50_ (Table 1), although additional coverage and efficacy were demonstrated [139]. Another ACE2 affinity matured molecule called sACE2_2_.v2.4-8h, which contained T27Y, L79T, and N330Y mutations, exhibited a greater than 100-fold increase in neutralization as a soluble ACE2 [124], but the improvement for its bivalent IgG fusion sACE2_2_.v2.4-IgG1 [132,133] was not reported and only about an 18-fold enhancement was achieved for another IgG fusion, ACE2.1mb, which contained the same three affinity mutations [129] (Table 1). Overall, these constructs with or without affinity mutations have not achieved lower than 10ng/mL of IC_50_ for neutralization (Table 1). Further, it should be noted that the ACE2 mutations do not always improve the antiviral potency [125,138]. Most importantly, these molecules, such as 3N39v4, could even lose sensitivity to certain viruses as compared to their WT control [138]. Therefore, whether these non-native ACE2 formats can withstand the RBD mutations found in the recently emerged SARS-CoV-2 Omicron viruses and variants/subvariants to emerge in the future is still an open question.

Another method to improve the potency of the ACE2 trap is ACE2 multimerization, in which multiple copies of ACE2 are assembled to increase valency for better antiviral activity. These ACE2 multimers include the early reported trimer and tetramer version of ACE2 [126,135,136], as well as the most recently developed pentamer and hexamer formats [102,142,143,144]. The trimer and tetramer ACE2s, such as ACE2_615_-foldon and ACE2-Fc-TD, did show improved potency of over 10-fold in comparison to the corresponding bivalent controls, but the IC_50_s are still above 10 ng/mL (Table 1), suggesting there is still significant room for further improvement.

Previously, we researched the possibility of improving SARS-CoV-2 mAb-based therapeutics by taking advantage of the decameric structure of IgM. The variable regions of several anti-RBD mAbs were converted into matched recombinant IgG and IgM pairs. These two forms with identical variable regions were then tested for their ability to neutralize SARS-CoV-2. It was found that IgM Abs were significantly more potent than the matched IgGs for viral neutralization. For example, for the pair named IgM-14 and IgG-14, the IgM-14 was 480-fold more potent than the bivalent IgG due to enhanced affinity and avidity, and it cleared virus in vivo much better than the IgG-14 [145]. This finding has been recently confirmed and reported by other investigators [146,147,148]. Together, these data suggest that strategies that utilize higher valency molecules are likely to exhibit superior in vitro potency and in vivo efficacy. Based on this assumption, we reasoned that by using an IgM Fc core to create a molecule displaying ten copies of ACE2, we could generate a more potent and durable antiviral intervention. To this end, we have designed and constructed multimeric ACE2 candidates that display ten copies of ACE2 with or without the neck domain (dACE2 or mACE2) by fusing them to a truncated IgM Fc core scaffold composed of IgM Cµ3, Cµ4, and tailpiece, resulting in dACE2-IgM and mACE2-IgM pentamer, respectively [102], (Table 1). For pseudovirus neutralization, both dACE2 and mACE2-IgM exhibited single-digit ng/mL or pM IC_50_ against WT, major SARS-CoV-2 variants, and Omicron sublineages including BA.1, BA.2, BA.2.12.1, BA.2.75, BQ.1, BQ.1.1, XBB.1, and XBB.1.5 while the corresponding bivalent controls all had much lower activity [102]. For authentic virus neutralization against Omicron viruses, including BA.1 and BA.2, our ACE2-IgMs demonstrated about 10 pM neutralization IC_50_ values with about 300 to 500-fold higher potency than the bivalent controls on a molar basis [102]. In addition, these pentamer traps can also potently neutralize SARS-CoV-1 and human alpha CoV NL63 that could cause severe infections in infants and immunocompromised patients [149,150], and multiple animal CoVs with pandemic potential [102]. We further demonstrated the in vivo efficacy of mACE2-IgM in protecting against both lethal and mild SARS-CoV-2 infections in K18-hACE2 mice with a single intranasal dose of prophylactic or therapeutic treatment [102].

Similar constructs were prepared either as ACE2-IgM pentamer or hexamer by two other groups (Table 1), but there was a lack of demonstration of in-vivo protection against lethal challenge [142,143]. In addition, another study reported the IgM-like ACE2 composed of mixtures of predominant hexamer and pentamer, and minor fractions of tetramer, trimer, and dimer, which was generated via fusion with IgG Fc that carries the IgM tailpiece for multimerization (Table 1). This IgM-like ACE2, or HH-120, was shown to have similar potency and coverage as we described, and has been evaluated in preclinical and clinical studies [144] (NCT05116865, NCT06039163, NCT05753878, NCT05713318, NCT05787418). Together with the observation of enhanced RBD bindings, neutralizations, and in-vivo protections, we and others have demonstrated that the potency of ACE2-based virus trap can be dramatically enhanced with IgM multimerization, overcoming the low potency issue that previously plagued the early versions of ACE2 viral traps. Therefore, these ACE2 multimers represent one of the best solutions for addressing the immune evasion of SARS-CoV-2.

## 5. Conclusions

Although the SARS-CoV-2 pandemic was declared over, its evolution has not stopped, and COVID-19 is still one of the major respiratory infectious diseases that continues to cause morbidity and death in the US and the rest of the world. In fact, we have just started to appreciate the remarkable ability of SARS-CoV-2 to evolve and transmit, as illustrated by the unprecedented diverse evolution and immune evasion of the Omicron variant, which are highlighted by the lately emerged BA.2.86 and JN.1. However, regardless of the evolution and emergence of variants and sublineages, the interaction and binding of SARS-CoV-2 spike protein RBD with ACE2 receptor remains essential for viral entrance and infection. Instead of focusing on targeting viral RBD sequences that are subject to continual mutation, an alternative approach like an ACE2-IgM trap could provide a durable therapeutic solution for addressing the challenge of Omicron virus evolution that has continued for over two years, and other CoVs dependent on ACE2 as a major receptor for infection, which include alpha CoV that could cause lethal infection in certain populations, and animal-origin sarbecoviruses with pandemic potential.

## Figures and Tables

**Figure 1 viruses-16-00697-f001:**
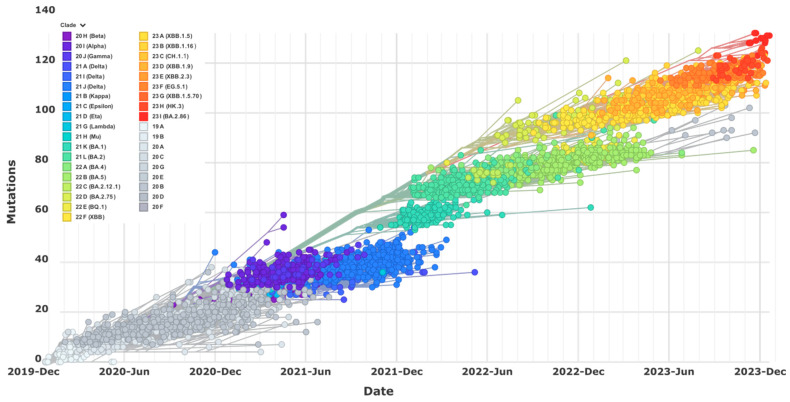
Clock tree of SARS-CoV-2 evolution. The phylogeny and accumulated mutations were plotted and observed for the 3728 globally representative genome samples of different SARS-CoV-2 variants and subvariants that have emerged since late 2019 (Accessed online at https://nextstrain.org/ on 30 December 2023).

**Figure 2 viruses-16-00697-f002:**
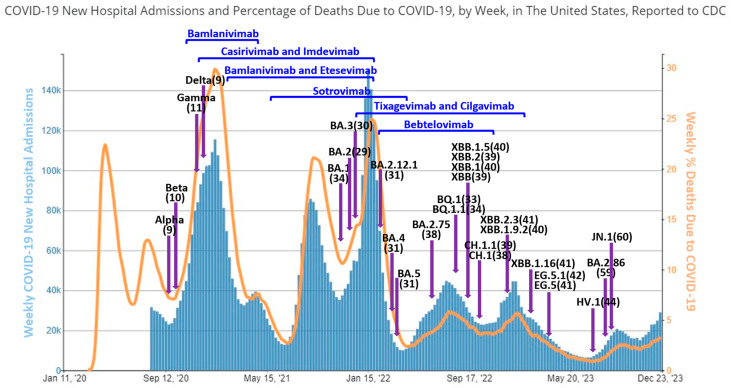
In-use duration of mAbs with the emergency use authorization (EUA) before and after the emergence of the Omicron variant and subvariants in association with hospital admissions and deaths in the US. The EUA approval and revoke dates were based on the FDA’s COVID-19 updates for the drugs and non-vaccine biological products. The earliest detection time of the major variants, including Alpha, Beta, Gamma, Delta, and Omicron (BA.1), was based on the reports from the original and review articles [31,32], while for Omicron subvariants, the earliest submission date for the samples deposited in GISAID database was used. The total number of spike protein mutations for each variant and subvariant was provided with continuous deletions counted as one. The trend chart for the weekly COVID-19 hospital admissions and deaths was downloaded from the CDC’s COVID Data Tracker website on 31 December 2023.

**Figure 3 viruses-16-00697-f003:**
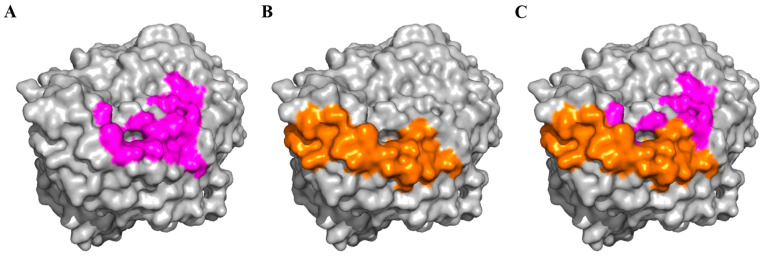
Footprints of CoV RBD binding on ACE2. The specific RBD residues that interact with ACE2 were mapped onto the ACE2 structure (PDB: 6M1D) for the alpha CoV NL63 (**A**) in magenta and Beta SARS-CoV-2 WT virus (**B**) in orange with overlay (**C**) based on the structures of PDB: 3KBH and 6VW1, respectively. The graphs were generated by PyMOL software 2.5.5.

**Table 1 viruses-16-00697-t001:** Summary of ACE2 traps developed for SARS-CoV-2 intervention.

Format	Name	Affinity Mutation	Length	Fc	K_D_ ^†^ (nM)	Neutralization		Reference
						IC_50_ ^‡^ (ng/mL)	Variants Tested	
Soluble ACE2	sACE2_2_(WT)-8h	N/A	740	N/A	ND	≈8700	ND	[124]
	sACE2_2_.v2.4-8h	T27Y, L79T, N330Y	740	N/A	ND	≈75	ND	[124]
	APN01	N/A	740	N/A	16.2	3200	Alpha, Beta, Gamma	[127,130,131]
	208e	N/A	740	N/A		2190	ND	[125]
	ACE2_615_	N/A	615	N/A	76.8	67,700	ND	[126]
	ACE2–1-618-DDC-ABD	N/A	618	N/A	ND	>45,000	Gamma, Delta	[128]
	sACE2	N/A	740	N/A	ND	3173	Alpha, Beta, Gamma, Delta, BA.1, BA.2	[129]
Bivalent ACE2	sACE2_2_(WT)-IgG1	N/A	740	IgG1	22	ND	ND	[124]
	sACE2_2_.v2.4-IgG1	T27Y, L79T, N330Y	740	IgG1	0.6	ND	BA.1, BA.2	[124,132,133]
	13	N/A	614	IgG1	10.8	430	ND	[125]
	208	N/A	740	IgG1	2.8	710	ND	[125]
	CVD019	K31F, H34I, E35Q	614	IgG1	0.89	310	ND	[125]
	CVD293	K31F, H34I, E35Q	740	IgG1	0.23	36	ND	[125]
	HLX71	N/A	740	IgG	40.1	5560	Alpha, Beta, Gamma, Iota, Epsilon, Delta,	[134]
	ACE2-Fc	615	IgG1	IgG1	21	456	Alpha, Beta	[135,136]
	ACE2m615-Fc	N/A	615	IgG	22.3	8400	ND	[126]
	WT	N/A	615	IgG1	17.6	24,800	ND	[137]
	3N39v2	A25V, K31N, E35K, L79F	615	IgG1	0.64	82	Alpha, Beta, Gamma	[137]
	ACE2(WT)-Fc	N/A	740	IgG1	ND	≈600	Alpha, Delta, BA.1	[138]
	3N39v4	A25V, K31N, E35K, T92Q	740	IgG1	ND	≈60	Alpha, Delta, BA.1, BA.2, BA.2.12.1, BA.2.75, BA.4/5, BQ.1, XBB	[138,139]
	MDR504 hACE2-Fc	N/A	740	IgG1	0.8	536	ND	[140]
	PD-CLD-Fc	N/A	740	IgG4	3.4	1.2 (nM)	Beta, BA.1	[141]
	PD-Fc		615		9.4	12 (nM)	ND	
	PD-Li-Fc		615		6.4	12 (nM)		
	Fc-Li-PD		615		3.1	3.1 (nM)		
	ACE2.mb	N/A	740	IgG1-CH3	ND	523	Alpha, Beta, Gamma, Delta, BA.1, BA.2	[129]
	ACE2.1mb	T27Y, L79T, N330Y	740	IgG1-CH3	ND	98	Alpha, Beta, Gamma, Delta, BA.1, BA.2	[129]
Multimer ACE2	ACE2_615_-foldon (trimer)	N/A	615	IgG	1.15	690	ND	[126]
	ACE2_615_-foldon mutant (trimer)	T27W	615	IgG	0.06	120	ND	[126]
	ACE2-Fc-TD (tetramer)	N/A	615	IgG1	3.9	36	Alpha, Beta	[135,136]
	ACE2-IgM-Fc (hexamer)	N/A	740	IgM without J chain	0.9	0.18 (nM)	BA.1	[142]
	IgM ACE2 (pentamer)	N/A	740	IgM with J chain	0.097	0.018 (nM)	Beta, Gamma, Delta, BA.1, BA.5	[143]
	HH-120 (IgM-like ACE2)	N/A	615	IgG plus IgM tail piece	<1 pM	9.6	Delta, BA.1, BA.1. BA.2, BA.2.12.1, BA.2.75, BA.2.76 BF.7, BA.4/5, BQ.1.1, XBB	[144]
	dACE2-IgM (pentamer)	N/A	740	IgM with J chain	<1 pM	5.4 (4.8 pM)	Alpha, Beta, Gamma, Delta, BA.1, BA.2, BA.2.12.1, BA.2.75, BA.4/5, BQ.1, BQ.1.1, XBB.1, XBB.1.5	[102]
	mACE2-IgM (pentamer)	N/A	615	IgM with J chain	<1 pM	5.0 (5.1 pM)	Alpha, Beta, Gamma, Delta, BA.1, BA.2, BA.2.12.1, BA.2.75, BA.4/5, BQ.1, BQ.1.1, XBB.1, XBB.1.5	[102]

Abbreviations: WT: wild-type, ND: not done. ^†^ K_D_: based on WT spike or RBD binding. ^‡^ IC_50_: based on WT (D614 or G614) SARS-CoV-2 pseudovirus or authentic neutralization.

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
