# Peer review of "SARS-CoV-2 Omicron: Viral Evolution, Immune Evasion, and Alternative Durable Therapeutic Strategies"

_viruses, 2024, doi:10.3390/v16050697_

Round 1
Reviewer 1 Report
Comments and Suggestions for Authors
Dear authors,
Your manuscript “SARS-CoV-2 Omicron: Viral Evolution, Immune Evasion, and Alternative Durable Therapeutic Strategies” makes the review of SARS-CoV-2 Omicron evolution, immune interaction and possible therapeutic strategies against it.
The manuscript is well written, contains all necessary parts and gives a lot of well systematized information to readers.
However, I have some remarks:
Figure 1 – What software did you use to build the clock tree? Needs to be specified here. If the clock tree was obtained from other source, it should be also stated.
Figure 3 – As far as I understand from the text, the data presented are your own experiments. So, it should be described or made a reference to a work describing the approaches for such analysis.
Author Response
Dear Reviewer,
Attached please find our response to the comments and suggestions in blue.

Reviewer 2 Report
Comments and Suggestions for Authors
Understanding how viruses such as SARS-CoV-2 evolve and evade both therapeutics and vaccines is key to the discovery of effective countermeasures. Quick SARS-CoV-2 changes due to many environmental impacts have made developing long-lasting Ab therapeutics difficult and many Ab products had only short life. Because of these severe changes in the virus, most companies have abandoned developing new Ab treatments against SARS-CoV-2. In this review “SARS-CoV-2 Omicron: Viral Evolution, Immune Evasion, and Alternative Durable Therapeutic Strategies” Hailong Guo and colleagues put forward an alternative way to approach SARS-CoV-2 therapeutics. First, the authors go through the evaluation of the SARS-CoV-2 till last December 2023. In Fig 2, the authors show how different mAbs with emergency use authorization were employed for a short time against the Omicron variant and subvariants. Then the authors make a case for using ACE2 as the trap which may be used against a variety of SARS-CoVs. The authors map out molecular interactions of ACE2 with many CoVs and show many similarities in the interactions. The review finishes with observations that multimeric ACE2 on Fc backbone has far more efficient binding than native ACE2. The authors have published on using IgM-ACE2 as therapeutics against multiple CoVs and they believe that this mode of therapeutics will overcome variant changes.
Overall, this is a well-written manuscript and timely. I only have a few minor suggestions.
1 1. The authors need to discuss both small and large molecules.
2 2. The authors neglected to mention targets besides Spike such as the polymerases.
3 3. The reviewer believes the use of Durable in the title is a bit confusing. The title should be changed to …….Alternative Broad-Spectrum Therapeutic Strategies
Author Response

(The authors gave the same response as above.)

Reviewer 3 Report
Comments and Suggestions for Authors
The review highlights the remarkable evolutionary adaptability of the SARS-CoV-2 Omicron variant, emphasizing the importance of targeting the ACE2 receptor rather than the virus’s RBD, which is susceptible to changes during viral evolution. It provides a detailed and non-traditional therapeutic perspective in the field of SARS-CoV-2 research. One concern is that the authors do not extensively elaborate on ACE2 therapeutics in the context of the evolutionarily diverse Omicron variant. However, the overall flow of the review is good, making it a pleasurable read. I have some minor comments below:
Line 74: BA.1, BA.2 and BA.2 or BA.3?
Line 203: subvariant was not be able to complete…. Please correct the typo.
Line 252: It should be evolutionarily instead of evolutionally
Author Response

(The authors gave the same response as above.)

Round 2
Reviewer 1 Report
Comments and Suggestions for Authors
Dear authors,
Thank you very much for your revisions made.
Now I have no remarks.